# Universal Guidance for Diffusion Models

**Arpit Bansal\***
University of Maryland
bansal01@umd.edu

**Hong-Min Chu\***
University of Maryland

**Avi Schwarzschild**
University of Maryland

**Soumyadip Sengupta**
University of North Carolina

**Micah Goldblum**
New York University

**Jonas Geiping**
University of Maryland

**Tom Goldstein**
University of Maryland

## Abstract

Typical diffusion models are trained to accept a particular form of conditioning, most commonly text, and cannot be conditioned on other modalities without retraining. In this work, we propose a universal guidance algorithm that enables diffusion models to be controlled by arbitrary guidance modalities without the need to retrain any use-specific components. We show that our algorithm successfully generates quality images with guidance functions including segmentation, face recognition, object detection, and classifier signals. Code is available at github.com/arpitbansal297/Universal-Guided-Diffusion.

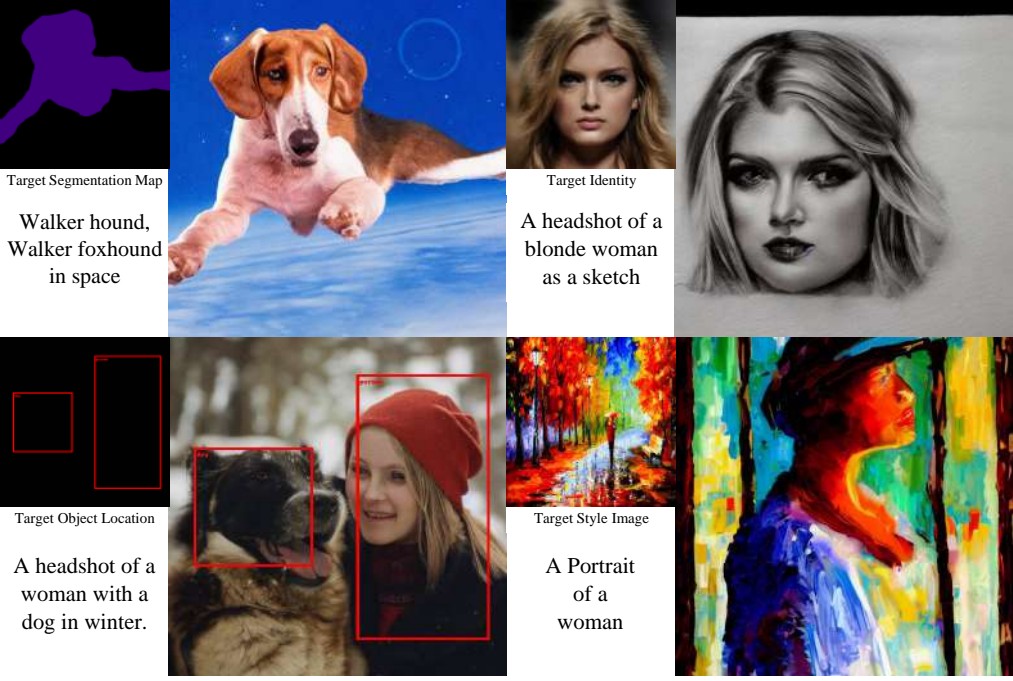

Figure 1: Diffusion guided by off-the-shelf networks. Top left: segmentation guidance, top-right: face recognition guidance, bottom-left: object detection guidance, bottom-right: style-transfer.

# 1 INTRODUCTION

Diffusion models are powerful tools for creating digital art and graphics. Much of their success stems from our ability to carefully control their outputs, customizing results for each user's individual needs. Most models today are controlled through *conditioning*. With conditioning, the diffusion model is built from the ground up to accept a particular modality of input from the user, be it descriptive text, segmentation maps, class labels, etc. While conditioning is a powerful tool, it results in models that are handcuffed to a single conditioning modality. If another modality is needed, a new model has to be trained, often from scratch. Unfortunately, the high cost of training makes this prohibitive for most users.

A more flexible approach to controlling model outputs is to use *guidance*. In this approach, the diffusion model acts as a generic image generator, and is not required to understand a user's instructions. The user pairs this model with a guidance function that measures whether some criterion has been met. For example, one could guide the model to minimize the CLIP score between the generated image and a text description of the user's choice. During each iteration of image creation, the iterates are nudged down the gradient of the guidance function, causing the final generated image to satisfy the user's criterion.

In this paper, we study guidance methods that enable any off-the-shelf model or loss function to be used as guidance for diffusion. Because guidance functions can be used without re-training or modification, this form of guidance is *universal* in that it enables a diffusion model to be adapted for nearly any purpose.

From a user perspective, guidance is superior to conditioning, as a *single* diffusion network is treated like a foundational model that provides universal coverage across many use cases, both commonplace and bespoke. Unfortunately, it is widely believed that this approach is infeasible. While early diffusion models relied on classifier guidance (Dhariwal & Nichol, 2021), the community quickly turned to classifier-free schemes (Ho & Salimans, 2022) that require a model to be trained from scratch on class labels with a particular frozen ontology that cannot be changed (Nichol et al., 2021; Rombach et al., 2022; Bansal et al., 2022).

The difficulty of using guidance stems from the domain shift between the noisy images used by the diffusion sampling process and the clean images on which the guidance models are trained. When this gap is closed, guidance can be performed successfully. For example, Nichol et al. (2021) successfully use a CLIP model as guidance, but only after re-training CLIP from scratch using noisy inputs. Noisy retraining closes the domain gap, but at a very high financial and engineering cost. To avoid the additional cost, we study methods for closing this gap by changing the sampling scheme, rather than the model.

To this end, our contributions are summarized as follows:

- We propose an algorithm that enables universal guidance for diffusion models. Our proposed sampler evaluates the guidance models only on denoised images, rather than noisy latent states. By doing so, we close the domain gap that has plagued standard guidance methods. This strategy provides the end-user with the flexibility to work with a wide range of guidance modalities and even multiple modalities simultaneously. The underlying diffusion model remains fixed and no fine-tuning of any kind is necessary.

- We demonstrate the effectiveness of our approach for a variety of different constraints such as *classifier labels*, *human identities*, *segmentation maps*, *annotations from object detectors*, and constraints arising from *inverse linear problems*.

# 2 BACKGROUND

We first briefly review the recent literature on the core framework behind diffusion models. Then, we define the problem setting of controlled image generation and discuss previous related works.

## 2.1 DIFFUSION MODELS

Diffusion models are strong generative models that proved powerful even when first introduced for image generation (Song & Ermon, 2019; Ho et al., 2020). The approach has been successfully extended to a number of domains, such as audio and text generation (Kong et al., 2020; Huang et al., 2022; Austin et al., 2021; Li et al., 2022).

We introduce (unconditional) diffusion formally, as it is helpful in describing the nuances of different types of models. A diffusion model is defined as a combination of a $T$-step forward process and a $T$-step reverse process. Conceptually, the forward process gradually adds Gaussian noise of different magnitudes to a clean data point $z_0$, while the reverse process attempts to gradually denoise a noisy input in hopes of recovering a clean data point. More concretely, given an array of scalars representing noise scales $\{\alpha_t\}_{t=1}^T$ and an initial, clean data point $z_0$, applying $t$ steps of the forward process to $z_0$ yields a noisy data point

$$z_t = \sqrt{\alpha_t}z_0 + (\sqrt{1 - \alpha_t})\epsilon, \ \epsilon \sim \mathcal{N}(0, \mathbf{I}). \tag{1}$$

A diffusion model is a learned denoising network $\epsilon_\theta$. It is trained so that for any pair $(z_0, t)$ and any sample of $\epsilon$,

$$\epsilon_\theta(z_t, t) \approx \epsilon = \frac{z_t - \sqrt{\alpha_t}z_0}{\sqrt{1 - \alpha_t}}. \tag{2}$$

The reverse process takes the form $q(z_{t-1}|z_t, z_0)$ with various detail definitions, where $q(\cdot|\cdot)$ is generally parameterized as a Gaussian distribution. Different works also studied different approximations of the unknown $q(z_{t-1}|z_t, z_0)$ used to perform sampling. For example, denoising diffusion implicit model (DDIM) (Song et al., 2021a) first computed a *predicted* clean data point

$$\hat{z}_0 = \frac{z_t - (\sqrt{1 - \alpha_t})\epsilon_\theta(z_t, t)}{\sqrt{\alpha_t}}, \tag{3}$$

and sample $z_{t-1}$ from $q(z_{t-1}|z_t, \hat{z}_0)$ by replacing unknown $z_0$ with $\hat{z}_0$. On the other hand, while the details of individual sampling methods vary, all sampling methods produce $z_{t-1}$ based on current sample $z_t$, current time step $t$ and a predicted noise $\hat{\epsilon}$. To ease the notation burden, we define a function $S(\cdot, \cdot, \cdot)$ as an abstraction of the sampling method, where $z_{t-1} = S(z_t, \hat{\epsilon}, t)$.

## 2.2 CONTROLLED IMAGE GENERATION

In this paper, we focus on controlled image generation with various constraints. Consider a differentiable guidance function $f$, for example a CLIP feature extractor or a segmentation network. When applied to an image, we obtain a vector $c = f(x)$. We also consider a function $\ell(\cdot, \cdot)$ that measures the closeness of two vectors $c$ and $c'$. Given a particular choice of $c$, which we call a *prompt*, the corresponding constraint (based on $c, \ell$, and $f$) is formalized as $\ell(c, f(z)) \approx 0$, and we aim to generate a sample $z$ from the image distribution satisfying the constraint. In plain words, we want to generate an in-distribution image that matches the prompt.

Prior work on controlled generative diffusion falls into two main categories. We refer to the first category as conditional image generation, and the second category as guided image generation. Next, we discuss the characteristics of each category and better situate our work among existing methods.

**Conditional Image Generation.** Methods from this category require training new diffusion models that accept the prompt as an additional input (Ho & Salimans, 2022; Bansal et al., 2022; Nichol et al., 2021; Whang et al., 2022; Wang et al., 2022a; Li et al., 2023; Zhang & Agrawala, 2023). For example, Ho & Salimans (2022) proposed classifier-free guidance using class labels as prompts, and trained a diffusion model by linear interpolation between unconditional and conditional outputs of the denoising networks. Bansal et al. (2022) studied the case where the guidance function is a known linear degradation operator, and trained a conditional model to solve linear inverse problems. Nichol et al. (2021) further extended classifier-free guidance to text-conditional image generation with descriptive phrases as prompts, and trained a diffusion model to enforce the similarity between the CLIP (Radford et al., 2021) representations of the generated images and the text prompts. These methods are successful across different types of constraints, however the requirement to retrain the diffusion model makes them computationally intensive.

**Guided Image Generation.** Works in this category employed a frozen pre-trained diffusion model as a foundation model, but modify the sampling method to guide the image generation with feedback from the guidance function. Our method falls into this category. Prior work that studied guided image generation did so with a variety of restrictions and external guidance functions (Dhariwal & Nichol, 2021; Kawar et al., 2022; Wang et al., 2022b; Chung et al., 2022a; Lugmayr et al., 2022; Chung et al., 2022b; Graikos et al., 2022). For example, Dhariwal & Nichol (2021) proposed classifier guidance, where they trained a classifier on images of different noise scales as the guidance function $f$, and included gradients of the classifier during the sampling process. However, a classifier for noisy images is domain-specific and generally not readily available – an issue our method circumvents. Wang et al. (2022b) assumed the external guidance functions to be linear operators, and generated the component of images residing in the null space of linear operators with the foundation model. Unfortunately, extending that method to handle non-linear guidance functions is non-trivial. Chung et al. (2022a) studied general guidance functions, and modified the sampling process with the gradient of guidance function calculated on the expected denoised images. Nevertheless, the authors only presented results with simpler non-linear guidance functions such as non-linear blurring.

In this work, we study universal guidance algorithms for guided image generation with diffusion models using any off-the-shelf guidance functions $f$, such as object detection or segmentation networks.

## 3 Universal Guidance

We propose a guidance algorithm that augments the image sampling method of a diffusion model to include guidance from an off-the-shelf auxiliary network. Our algorithm is motivated by an empirical observation that the reconstructed clean image $\hat{z}_0$ obtained by Eq. (3), while naturally imperfect, is still appropriate for a generic guidance function to provide informative feedback to guide the image generation. In Sec. 3.1, we motivate our *forward universal guidance* by extending classifier guidance Dhariwal & Nichol (2021) to leverage this observation and handle generic guidance functions. In Sec. 3.2, we propose a supplementary *backward universal guidance* to help enforce the generated image to satisfy the constraint based on the guidance function $f$. In Sec. 3.3, we discuss a simple yet helpful stepwise refinement trick to empirically improve the fidelity of generated images.

### 3.1 Forward Universal Guidance

To guide the generation with information from the external guidance function $f$ and the loss function $\ell$, an immediate thought is to extend classifier guidance (Dhariwal & Nichol, 2021) to accept any general guidance function. Concretely, given a class prompt $c$, classifier guidance performs classification-guided sampling by replacing $\epsilon_\theta(z_t, t)$ in each sampling step $S(z_t, t)$ with

$$\hat{\epsilon}_\theta(z_t, t) = \epsilon_\theta(z_t, t) - \sqrt{1 - \alpha_t} \nabla_{z_t} \log p(c|z_t). \tag{4}$$

Defining $\ell_{ce}(\cdot, \cdot)$ to be the cross-entropy loss and $f_{cl}$ to be the guidance function that outputs classification probability, Eq. (4) can be re-written as

$$\hat{\epsilon}_\theta(z_t, t) = \epsilon_\theta(z_t, t) + \sqrt{1 - \alpha_t} \nabla_{z_t} \ell_{ce}(c, f_{cl}(z_t)). \tag{5}$$

However, directly replacing $f_{cl}$ and $\ell_{ce}$ with any off-the-shelf guidance and loss functions does not work in practice, as $f$ is most likely trained on clean images and fails to provide meaningful guidance when the input is noisy.

To address the issue, we observe that

$$p(c|z_t) = \int p(c|z_0, z_t) p(z_0|z_t) dz_0 = \mathbb{E}_{z_0 \sim p(z_0|z_t)}[p(c|z_0)]. \tag{6}$$

where $c$ is conditionally-independent with $z_t$ given $z_0$. Leveraging the fact that we can obtain a *predicted* clean image $\hat{z}_0$ by Eq. (3) with $\epsilon_\theta(z_t, t)$, we approximate the expectation in Eq. (6) as $\mathbb{E}_{z_0 \sim p(z_0|z_t)}[p(c|z_0)] \approx p(c|\hat{z}_0)$. This leads to our proposed guided sampling procedure

$$\hat{\epsilon}_\theta(z_t, t) = \epsilon_\theta(z_t, t) + s(t) \cdot \nabla_{z_t} \ell(c, f(\hat{z}_0)) \tag{7}$$

where $s(t)$ controls the guidance strength for each sampling step and

$$\nabla_{z_t} \ell(c, f(\hat{z}_0)) = \nabla_{z_t} \ell \left( c, f \left( \frac{z_t - \sqrt{1 - \alpha_t} \epsilon_\theta(z_t, t)}{\sqrt{\alpha_t}} \right) \right)$$

as in Eq. (3). We term Eq. (7) forward universal guidance, or forward guidance in short. In practice, applying forward guidance effectively brings the generated image closer to the prompt while keeping the generation trajectory in the data manifold. We note that a related approach is also studied in Chung et al. (2022a), where the guidance step is computed based on $E[z_0|z_t]$. The approach drew inspiration from the score-based generative framework (Song et al., 2021b), but resulted in a different update method.

## 3.2 BACKWARD UNIVERSAL GUIDANCE

As will be shown in Sec. 4.2, we observe that forward guidance sometimes over-prioritizes maintaining the "realness" of the image, resulting in an unsatisfactory match with the given prompt. Simply increasing the guidance strength $s(t)$ is suboptimal, as this often results in instability as the image moves off the manifold faster than the denoiser can correct it.

---

**Algorithm 1** Universal Guidance

**Parameter:** Refinement steps $k$, backward guidance steps $m$, and guidance strength $s(t)$,
**Required:** $z_t$ the noisy vector at a given time-step $t$, diffusion model $\epsilon_\theta$, noise scale $\alpha_t$, guidance function $f$, loss function $\ell$, and prompt $c$
**for** $n = 1, 2, \ldots, k$ **do**
    Calculate $\hat{z}_0$ as in Eq. (3).
    Calculate $\hat{\epsilon}_\theta$ using forward universal guidance as in Eq. (7).
    **if** $m > 0$ **then**
        Calculate $\Delta z_0$ by minimizing Eq. (8) with $m$ steps of gradient descent.
        Perform backward universal guidance by
        $\hat{\epsilon}_\theta \leftarrow \hat{\epsilon}_\theta - \sqrt{\alpha_t/(1 - \alpha_t)} \Delta z_0$ (see Eq. (10)).
    **end if**
    $z_{t-1} \leftarrow S(z_t, \hat{\epsilon}_\theta, t)$.
    $\epsilon' \sim \mathcal{N}(0, I)$.
    $z_t \leftarrow \sqrt{\alpha_t/\alpha_{t-1}} z_{t-1} + \sqrt{1 - \alpha_t/\alpha_{t-1}} \epsilon'$.
**end for**
**Return** $z_{t-1}$

---

To address the issue, we propose backward universal guidance, or backward guidance in short, to supplement forward guidance and help enforce the generated image to satisfy the constraint. The key idea of backward guidance is to optimize for a clean image that best matches the prompt based on $\hat{z}_0$, and linearly translate the guided change back to the noisy image space at step $t$. Concretely, instead of directly calculating $\nabla_{z_t} \ell(c, f(\hat{z}_0))$, we compute a guided change $\Delta z_0$ in clean data space as

$$\Delta z_0 = \arg\min_\Delta \ell(c, f(\hat{z}_0 + \Delta)). \tag{8}$$

Empirically, we solve Eq. (8) with $m$-step gradient descent, where we use $\Delta = 0$ as a starting point. Since $\hat{z}_0 + \Delta z_0$ minimizes $\ell(c, f(z))$ directly, $\Delta z_0$ is the change in clean data space that best enforces the constraint. Then, we translate $\Delta z_0$ back to the noisy data space of $z_t$ by calculating the *guided denoising prediction* $\tilde{\epsilon}$ that satisfies

$$z_t = \sqrt{\alpha_t}(\hat{z}_0 + \Delta z_0) + \sqrt{1 - \alpha_t} \tilde{\epsilon}. \tag{9}$$

Reusing Eq. (3), we can rewrite $\tilde{\epsilon}$ as an augmentation to the original denoising prediction $\epsilon_\theta(z_t, t)$ by

$$\tilde{\epsilon} = \epsilon_\theta(z_t, t) - \sqrt{\alpha_t/(1 - \alpha_t)} \Delta z_0. \tag{10}$$

Comparing to forward guidance, backward guidance (as Eq. (10)) produces an optimized direction for the generated image to match the given prompt, and hence prioritizes enforcing the constraint. Furthermore, calculation of a gradient step for Eq. (8) is computationally cheaper than forward guidance (Eq. (7)), and we can therefore afford to solve Eq. (8) with multiple gradient steps, further improving the match with the given prompt.

We note that the names "forward" and "backward" are used analogously to the forward and backward Euler methods.

### 3.3 UNIVERSAL STEPWISE REFINEMENT

Upon applying our universal guidance to standard generation pipelines, we observe that in some instances, the images generated exhibit artifacts or unusual behaviors that differentiate them from natural images. Similar findings have been reported in (Lugmayr et al., 2022; Wang et al., 2022b), where linear guidance functions were explored. Although we attempted to enhance the realism by adjusting the guidance strength $s(t)$, finding an optimal balance that

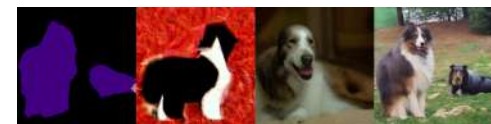

Figure 2: An example of how the Universal Stepwise Refinement (USR) helps segmentation-guided generation. The left-most figure is the given segmentation map, and the images generated with USR steps 1, 4 and 10 follow in order. Notice the increase of quality with USR steps.

both guarantees image realism and adherence to guidance constraints proved challenging, particularly with complex guidance functions. We hypothesize that the direction of guidance offered by our universal method might not always correlate with image realism, diverting the image from a natural image sampling trajectory.

Drawing motivation from Lugmayr et al. (2022); Wang et al. (2022b), we propose a more nuanced approach, termed **Universal Stepwise Refinement** (USR), to navigate these challenges. Specifically, after sampling $z_{t-1} = S(z_t, \hat{\epsilon}_t, t)$, we reintroduce Gaussian noise $\epsilon' \sim \mathcal{N}(0, \mathbf{I})$ to $z_{t-1}$, deriving $z'_t$ as per the equation:

$$z'_t = \sqrt{\alpha_t/\alpha_{t-1}} \cdot z_{t-1} + \sqrt{1 - \alpha_t/\alpha_{t-1}} \cdot \epsilon'. \tag{11}$$

Equation 11 ensures $z'_t$ retains the appropriate noise scale for input at time step $t$. Given a refinement step $k$, we repeat the Universal Stepwise Refinement $k$ times before advancing with the sampling for step $t-1$. This refined process enables the exploration of various regions of the data manifold at a consistent noise scale to reach a solution that aligns with both the guidance and image quality requisites. Our empirical assessments indicate that using USR mechanism on top of proper guidance strength $s(t)$ significantly enhanced the image realism while still maintaining the adherence to the prompt, as shown in Fig. 2.

We summarize our universal guidance algorithm composed of forward universal guidance, backward universal guidance and universal stepwise refinement for a single sampling step in Algorithm 1. For simplicity, the algorithm assumes only one guidance function, but can be easily adapted to handle multiple pairs of $(f, l)$. Additionally, the objectives of the forward and backward guidance do not have to be identical, allowing different ways to simultaneously utilize multiple guidance functions.

## 4 EXPERIMENTS

In this section, we present results testing our proposed universal guidance algorithm against a wide variety of guidance functions. Specifically, we experiment with Stable Diffusion (Rombach et al., 2022), a diffusion model that is able to perform text-conditional generation by accepting text prompt as additional input, and experiment with a purely unconditional diffusion model trained on ImageNet (Deng et al., 2009), where we use pre-trained model provided by OpenAI (Dhariwal & Nichol, 2021). We note that Stable Diffusion, while being a text-conditional generative model, can also perform unconditional image generation by simply using an empty string as the text prompt. We first present the experiment on Stable Diffusion for different guidance functions in Sec. 4.1, and present the results on ImageNet diffusion model in Sec. 4.2. Hyper-parameters used for different guidance functions, further ablation studies and selection procedures for suitable guidance strength $s(t)$ and refinement step $k$ can be found in the appendix.

### 4.1 RESULTS FOR STABLE DIFFUSION

In this section, we present the results of guided image generation using Stable Diffusion as the foundation model. The guidance functions we experiment with include a segmentation network, a face recognition network, an object detection network and style guidance with CLIP feature extractor (Radford et al., 2021). For experiments on Stable Diffusion, we discover that applying forward guidance already produce high-quality images that match the given prompt, and hence set $m = 0$. To perform forward guidance on Stable Diffusion, we forward the predicted clean latent variable computed by Eq. (3) through the image decoder of Stable Diffusion to obtain predicted clean

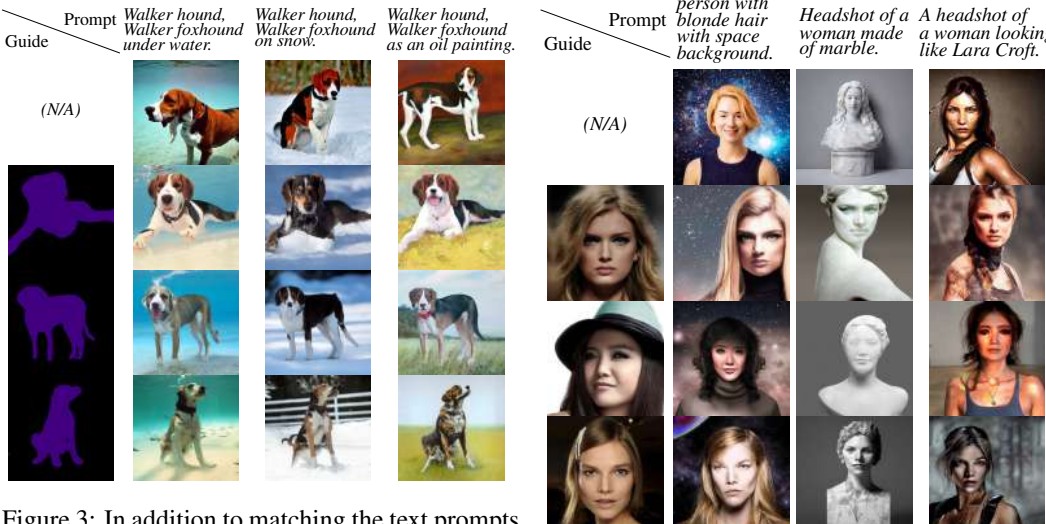

Figure 3: In addition to matching the text prompts (above each column), these images are guided by an image segmentation pipeline. Each column contains examples of images generated to match the prompt and the segmentation map in the left-most column. The top-most row contains examples generated without guidance.

Figure 4: Each column contains examples of images generated to match the prompt and the identity of the images in the left-most column. The top-most row contains examples generated without guidance.

images. We discuss the qualitative results and implementation details for each guidance function in its corresponding subsection. We summarize our quantitative evaluation in Tab. 1, where we evaluate how well the generated images match the external guidance with task-specific metrics, and include the similarity of CLIP embedding between text prompts and images generated with and without guidance. For all tasks, the minimum difference between CLIP similarities indicates that our algorithm performs guidance without sacrificing the effectiveness of the underlying diffusion model. We also include additional quantitative and qualitative results for text-guided generation in the appendix.

**Segmentation Map Guidance.** To guide image generation using a segmentation map as prompt, we use a MobileNetV3-Large (Howard et al., 2019) with a segmentation head, and a publicly available pre-trained model in PyTorch (Paszke et al., 2019). We use standard per-pixel cross-entropy loss between a given prompt and the predicted segmentation of generated images as our loss function $\ell$.

| Task | Metric | Value |
|------|--------|-------|
| Segmentation | mIoU | 0.898 |
|  | CLIP similarity | (0.247) 0.249 |
| Face | Face similarity | 0.642 |
|  | CLIP similarity | (0.287) 0.234 |
| Object detection | mAP@50 | 0.634 |
|  | CLIP similarity | (0.263) 0.246 |

Table 1: Quantitative analysis of different guidance applied on Stable Diffusion. The reference value in parenthesis is obtained with no external guidance.

In our experiment, we combine segmentation maps that depict objects of different shapes with new text prompts. We use the text prompt as a fixed additional input to Stable Diffusion to perform text-conditional sampling, and guide the text-conditional generated images to match the given segmentation maps. Qualitative results are presented in Fig. 3. From Fig. 3, we see that the generated images show a clear separation between object and background that matches the given segmentation map nearly perfectly. The generated object and background also each match their descriptive text (i.e. dog breed and environment description). Furthermore, the generated images are overall highly realistic. In Tab. 1, we evaluate mIoU between the ground truth segmentation map and the predicted segmentation of generated images to assess the match with constraint.

**Face Recognition Guidance.** For guiding generation to include a specific person's likeness, we propose combining face detection and facial recognition modules into one guidance function. This setup produces a facial attribute embedding from an image of a face. We use multi-task cascaded convolutional networks (MTCNN) (Zhang et al., 2016) for face detection, and we use facenet (Schroff et al., 2015) for facial recognition. The guidance function $f$ then crops out the detected face and

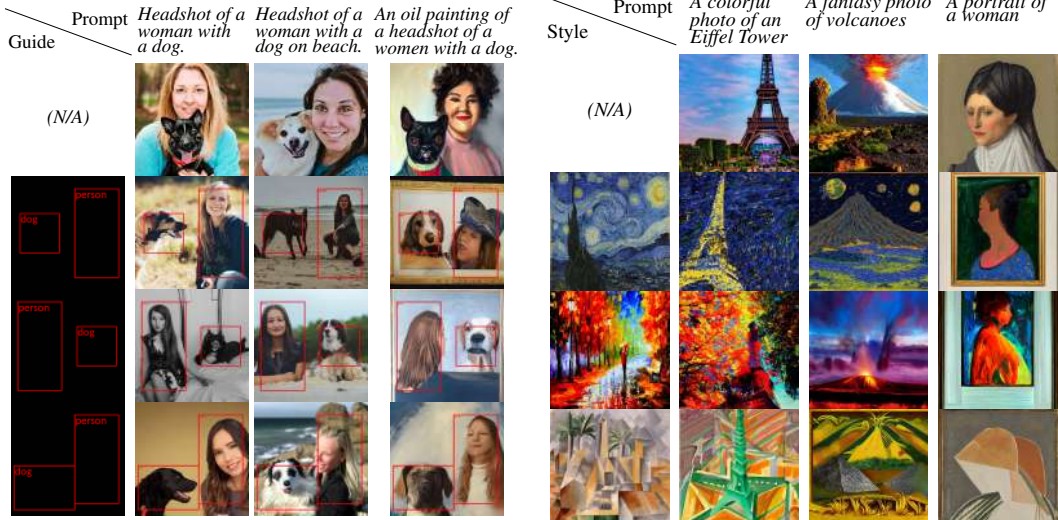

Figure 5: In addition to text prompts (above each column), these images are guided by an object detector. Each column contains examples of images generated to match the prompt and the bounding boxes used for guidance.

Figure 6: In addition to text prompts, these images are guided by a style image. Each column contains examples of images generated to match the text prompt and the style used for guidance.

outputs a facial attribute embedding as a prompt and we use an $l_1$-loss between embeddings as the loss function $\ell$.

We explore different combinations of face guidance and text prompts. Similarly to the segmentation case, we use the text prompt as a fixed additional conditioning to Stable Diffusion and guide this text-conditional trajectory with our algorithm so that the face in the generated image looks similar to the face prompt. In Fig. 4, we clearly see that the facial characteristics of a given face prompt are reproduced almost perfectly on the generated images. The descriptive text of either background, material, or style is also realized correctly and blends nicely with the generated faces. We again summarize our quantitative evaluation in Tab. 1. We evaluate the similarity between facial attributes of ground-truth identity and the generated faces. In general, two faces are considered to be from the same person if the similarity is over $0.5$, and our algorithm can effectively guide the generated face to meet the criteria.

**Object Location Guidance**    For Stable Diffusion, we also present the results of guided image generation with an object detection network. For this experiment, we use Faster-RCNN (Ren et al., 2015) with Resnet-50-FPN backbone (Li et al., 2021), a publicly available pre-trained model in Pytorch, as our object detector. We use bounding boxes with class labels as our object location prompt. We construct a loss function $\ell$ by the sum of three individual losses, namely (1) anchor classification loss, (2) bounding box regression loss and (3) region label classification loss.

We again experiment with different combinations of text prompt and object location prompt, and similarly use the text prompt as a fixed conditioning to Stable Diffusion. Using our proposed guidance algorithm, we perform guided image generation that generates and matches the objects presented in the text prompt to the given object locations. The results are presented in Fig. 5. We observe from Fig. 5 that objects in the descriptive text all appear in the designated location with the appropriate size indicated by the given bounding boxes. Each location is filled with appropriate, high-quality generations that align with varied image content prompts, ranging from "beach" to "oil painting". In Tab. 1, we use mAP@50 to measure how well the generated images satisfy the constraint.

**Style Guidance**    Finally, we conclude our experiments on Stable Diffusion by guiding the image generation based on a reference style given by a style image. To achieve so, we capture the reference style from the style image by the image feature extractor from CLIP, and use the resulting image embedding as prompts. The loss function calculates the negative cosine similarity between the embedding of generated images and the embedding of the style image. Similar to previous experiments,

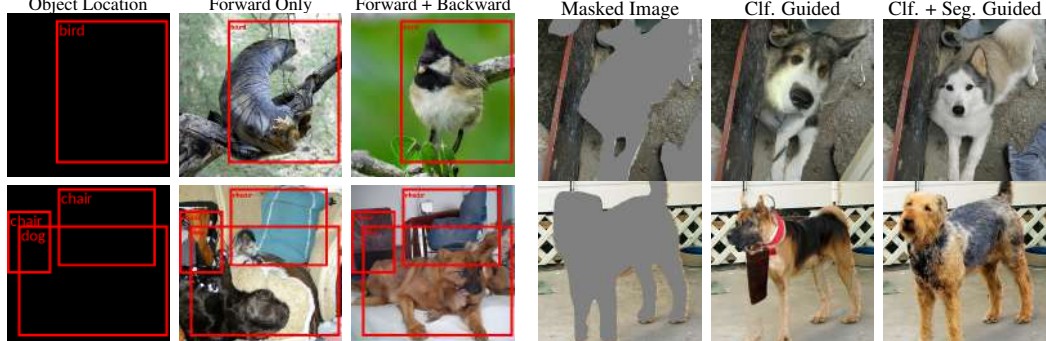

| Object Location | Forward Only | Forward + Backward | Masked Image | Clf. Guided | Clf. + Seg. Guided |

Figure 7: Generation guided by object detection with the unconditional ImageNet model. While both forward and backward guidance produces realistic images with the desired objects in the designated locations, forward guidance alone produces the wrong objects or the wrong locations/sizes.

Figure 8: Our guidance algorithm can incorporate feedback from multiple guidance functions. Left to right: The inpainting prompt; The classifier-guided inpainting output; Images generated with both classifier and segmentation guidance, where realistic dogs are generated exactly on the mask.

we control the content using text input as additional conditioning to the Stable Diffusion model. We experiment with combinations of different style images and different text prompts, and present the results in Fig. 6. From Fig. 6, we can see that the generated images contain contents that match the given text prompts, while exhibiting style that matches the given style images.

## 4.2 RESULTS FOR IMAGENET DIFFUSION

In this section, we present results for guided image generation using an unconditional diffusion model trained on ImageNet. We experiment with object location guidance and a hybrid guided image generation task which we term segmentation-guided inpainting. We also include additional experiments for CLIP guidance in the appendix. We will discuss results of each guidance separately.

**Object Location Guidance.** Similar to object location guidance for Stable Diffusion, we also use the same network architecture and the same pre-trained model as our object detection network, and construct an identical loss function $\ell$ for our guidance algorithm. However, unlike Stable Diffusion, object locations are the only prompts available for guided image generation. We experiment with different object location prompts using either (1) only forward universal

| Object Location | Fwd. Only | Fwd. + Bkd. |
|---|---|---|
| Bounding box 1 | 0.39 | **0.90** |
| Bounding box 2 | 0.18 | **0.36** |

Table 2: Quantitative analysis of forward guidance only versus combination of forward and backward guidance for object detection guidance on ImageNet with bounding boxes in Fig. 7. The metric is mAP@50.

guidance and (2) both forward and backward universal guidance. We observe from Fig. 7 that applying both forward and backward guidance generates images that are realistic and the objects match the prompt nicely. On the other hand, while images generated using only forward guidance remain realistic, they feature objects with mismatching categories and locations. The observation is further backed by quantitative evaluation presented in Tab. 2. The evaluation is based on mAP@50 between the ground truth object locations and the predicted bounding boxes of generated images, and clearly shows that the combination of forward and backward guidance leads to a much better match with the constraint. The results demonstrate the effectiveness of our universal guidance algorithm, and also validate the necessity of our backward guidance.

**Segmentation-Guided Inpainting.** In this experiment, we aim to explore the ability of our algorithm to handle multiple guidance functions. We perform guided image generation with combined guidance from an inpainting mask, a classifier and a segmentation network. We first generate images with masked regions as the prompt for inpainting. We then pick an object class $c$ as the prompt for classification and generate a segmentation mask where the masked regions are considered foreground objects of the same class $c$. We use $\ell_2$ loss on the non-masked region as the loss function for inpainting, and set the corresponding $s(t) = 0$, or equivalently only use backward guidance for inpainting. We use the same segmentation network as described in Sec. 4.1. For classification

guidance, we use the classifier that accepts noisy input (Dhariwal & Nichol, 2021), and perform the original classifier guidance as Eq. (4) instead of our forward guidance. The results in Fig. 8 show that when using both inpainting and classifier as guidance, our algorithm generates realistic images that both match the inpainting prompt and are classified correctly to the given object class. Adding in segmentation guidance, our algorithm further improves by making a near-perfect match to both the segmentation map and the inpainting prompt while maintaining realism, demonstrating that our algorithm effectively combines feedback from multiple guidance functions.

## 5   CONCLUSION

We propose a universal guidance algorithm that is able to perform guided image generation with any off-the-shelf guidance function based on a fixed foundation diffusion model. Our algorithm only needs guidance and loss functions to be differentiable, and avoids any retraining to adapt the guidance function or the foundation model to a specific type of prompt. We demonstrate promising results with our algorithm on complex guidance including segmentation, face recognition and object detection systems – and multiple guidance functions can even be used together.

## 6   ACKNOWLEDGEMENTS

This work was made possible by the ONR MURI program and the AFOSR MURI program. Commercial support was provided by Capital One Bank, the Amazon Research Award program, and Open Philanthropy. Further support was provided by the National Science Foundation (IIS-2212182), and by the NSF TRAILS Institute (2229885).

## REPRODUCIBILITY STATEMENT

we describe guidance functions and foundation diffusion models for experiments presented in Sec. 4 in the corresponding subsections. Hyperparameters for experiments described in Sec. 4 of the main paper can be found in Sec. B in the appendix. We also include the source code used to conduct the experiments described in the paper in our supplementary material.

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

# A    LIMITATIONS

Generation using universal guidance is typically slower than standard conditional generation for several reasons. Empirically, more than one iteration of denoising is required for the given noise level $t$ to generate high-quality images with complex guidance functions. However, the time complexity of our algorithm scales linearly with the number of refinement steps $k$, which slows down image generation when $k$ is large. That being said, for the applications presented in this paper, we were successful in generating images with smaller values of $k$. Also, as demonstrated in the main paper, backward guidance is required in certain scenarios to help generate images that match the given constraint. Computing backward guidance requires performing minimization with a multi-step gradient descent inner loop. While proper choices of gradient-based optimization algorithms and learning rate schedules significantly speed up the convergence of minimization, the time it takes to compute backward guidance inevitably becomes longer when the guidance function is itself a very-large neural network. Finally, we note that, to get optimal results, sampling hyper-parameters must be chosen individually for each guidance network.

# B    HYPER-PARAMETERS

In this section, we present the hyper-parameters for the different guidance functions i.e. face, segmentation, object location, and style guidance. We present the hyperparameters for experiments on Stable Diffusion in Sec. 4.1 in the Tab. 3, where we include coefficient $s_0$ to compute $s_t = s_0\sqrt{1-\alpha_t}$ and the number of Universal Stepwise Refinement ($k$). We also provide hyperparameters for experiments on ImageNet in Sec. 4.2 in Tab. 4

| Guidance | $s_0$ | $k$ |
|---|---|---|
| Face | 20000 | 2 |
| Object Location | 100 | 3 |
| Style Transfer | 6 | 6 |
| Segmentation | 400 | 10 |

Table 3: Hyper-parameters used in this paper for different guidance functions to reproduce the results for Stable Diffusion.

| Guidance | $s_0$ | $k$ |
|---|---|---|
| Object Location | 100 | 3 |
| Segmentation | 200 | 10 |

Table 4: Hyper-parameters used in this paper for different guidance functions to reproduce the results for ImageNet.

# C    CLIP GUIDANCE FOR STABLE DIFFUSION

CLIP (Radford et al., 2021) is a state-of-the-art text-to-image similarity model developed by OpenAI. We use the image feature extractor of CLIP to do text-guided image generation with our algorithm. We construct a loss function that calculates the negative cosine similarity between an image embedding and the CLIP text embedding produced by a given text prompt. We use $s(t) = 10\sqrt{1-\alpha_t}$ and $k = 8$ and use Stable Diffusion as an unconditional image generator.

We generate images guided by a number of text prompts. To further assess our universal guidance algorithm and compare guidance and conditioning, we also generate images using classical, text-conditional generation by Stable Diffusion with identical prompts as inputs, and summarize the results in Fig. 9. The results in Fig. 9 show that our algorithm can guide the generation to produce high-quality images that match the given text description, and are comparable with images generated by the specialized text-conditioning model. We also include qualitative results from experiments on DrawBench (Saharia et al., 2022). DrawBench is a widely-used and diverse list of text prompts. We randomly select 20 prompts and generate 10 images for each individual prompt. We compute CLIP

| Conditional Stable-Diffusion | Guided Stable-Diffusion |
|---|---|

*A photograph of an astronaut riding a horse.*

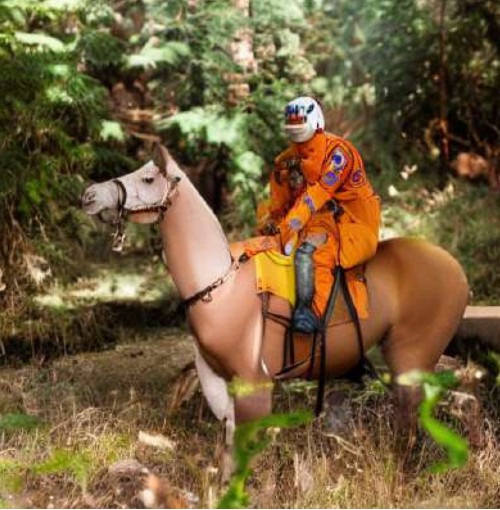

*An oil painting of a corgi wearing a party hat.*

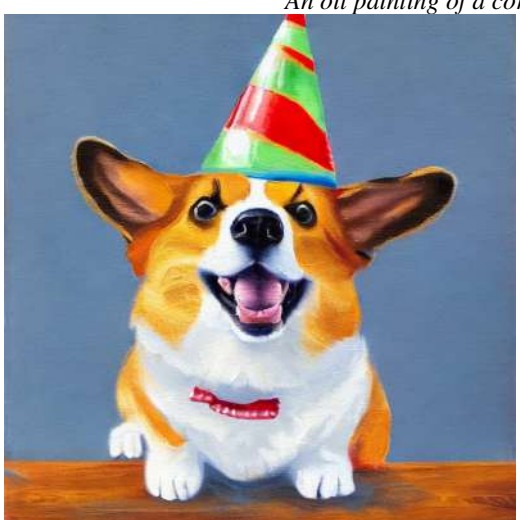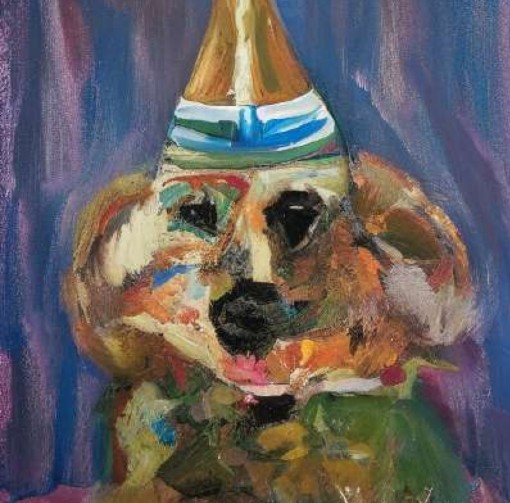

Figure 9: We compare the ability to match given text prompts between our universal guidance algorithm and text-conditional model trained from scratch. The results demonstrate that our universal algorithm is comparable to specialized conditional model on the ability to generate quality images that satisfy the text constraints.

score, the cosine similarity between CLIP feature of the text prompt and the associated generated image, on both Stable Diffusion and our algorithm, and report the average in Tab. 5. As demonstrated in the table, the performance of our algorithm is quantitatively similar to a dedicated text-conditional generator, while requiring no additional training at all for the underlying unconditional diffusion model.

|  | Stable Diffusion | Universal Guidance |
|---|---|---|
| CLIP Score | 0.2818 | 0.2632 |

Table 5: Quantitative results on DrawBench for Stable Diffusion and our algorithm.

# D CLIP GUIDANCE FOR IMAGENET DIFFUSION MODEL

| English foxhound by Edward Hopper | Van Gogh Style | Cake |

Figure 10: We show that unconditional diffusion models trained on ImageNet can be guided with CLIP to generate high-quality images that match the text prompts, even if these generated images should be *out of distribution*.

**CLIP Guidance.** We use the same construction of $f$ and $\ell$ for Stable Diffusion to perform CLIP-guided generation. We use only forward guidance for this experiment. To assess the limit of our universal guidance algorithm, we hand-crafted text prompts such that the matching images are *expected to be out of distribution*. In particular, our text prompts either designate art styles that are far from realistic or designate objects that do not belong to any possible class label of ImageNet. We present the results in Fig. 10, and from the results, we clearly see that our algorithm still successfully guides the generation to produce quality images that also match the text prompts. For all three images, we have $s(t) = w \cdot \sqrt{1 - \alpha_t}$, where $w$ is 2, 5 and 2 respectively and $k$ is 10, 5 and 10 respectively.

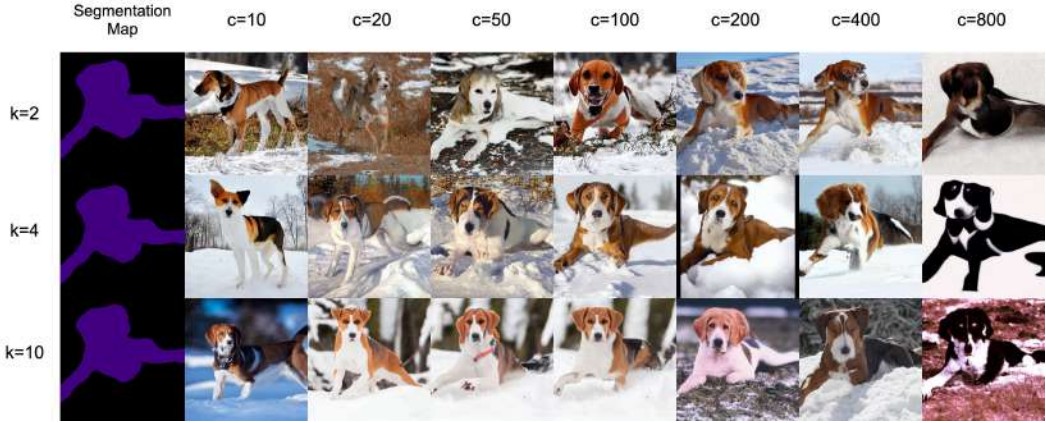

Figure 11: The qualitative results for effect of different guidance strength $s(t)$ and universal refinement step $k$ on segmentation guidance for Stable Diffusion. Here we use $s(t) = c \cdot \sqrt{1 - \alpha_t}$, and compare different $c$ instead.

# E  ABLATION STUDY AND PROCEDURE TO PICK HYPERPARAMETERS

In this section we present our ablation study on the effect of different parameters, namely guidance strength $s(t)$ and refinement step $k$, for our algorithm. We summarize the qualitative results in Fig. 11 on segmentation guidance for Stable Diffusion. From the figure, we observe that increasing $c$ alone leads to better matching between the segmentation map and the position of the generated dogs, but the quality of images also degrades, especially when $c$ is large ($c >= 200$). On the other hand, increasing $k$ noticeably improves the quality of images. For example, comparing the best image from $(k, c) = (4, 800)$ and $(k, c) = (10, 800)$, the latter clearly exhibits significantly better realness. We also quantitatively evaluate the effectiveness of the two parameters, and present the results in Tab. 6. We calculate the mIOU between the ground truth segmentation map and the segmentation map of generated images predicted by the given segmentation network, which assesses the match between generation and the given constraint. We also show cosine similarity between the clip attribute vectors of the generated images and the given text phrase. High cosine similarity indicates that generation guided with segmentation does not stray away from the text-conditional generation of the underlying diffusion model. In Tab. 6, we observe that when the step $m$ is fixed, increasing the guidance strength $c$ leads to a better match with external guidance. On the other hand, the clip similarity score is stable for $c$ in range of $(10, 200)$ before showing a sudden drop when $c >= 400$, indicating that it is more difficult for the underlying diffusion model to control the guided generation to match the text when $c$ is large. As for universal refinement step $k$, increasing $k$ generally improves both mIOU and clip similarity.

| Metric | Refinement step | c =10 | c = 20 | c = 50 | c = 100 | c =200 | c =400 | c =800 |
|---|---|---|---|---|---|---|---|---|
| | k=2 | 0.52 | 0.52 | 0.67 | 0.79 | 0.87 | 0.89 | 0.89 |
| mIoU | k=4 | 0.50 | 0.59 | 0.71 | 0.80 | 0.87 | 0.89 | 0.88 |
| | k=10 | 0.57 | 0.62 | 0.76 | 0.80 | 0.89 | 0.90 | 0.92 |
| | k=2 | 0.26 | 0.26 | 0.25 | 0.24 | 0.24 | 0.20 | 0.16 |
| CLIP sim. | k=4 | 0.26 | 0.27 | 0.26 | 0.25 | 0.25 | 0.21 | 0.14 |
| | k=10 | 0.24 | 0.25 | 0.26 | 0.26 | 0.25 | 0.22 | 0.18 |

Table 6: Quantitative evaluation for different combination of parameters of segmentation guidance on Stable Diffusion.

We also want to remark that the ablation study formalizes a principled way to pick suitable $k$ and $s(t)$ for different guidance functions. In particular, increasing $k$ is always beneficial to the generation quality, and the value of $k$ is limited only by the computational budget. Given a fixed $k$, there is generally a sweet spot for $s(t)$ that ensures both a match to the target and sufficient quality, and this sweet spot can be found with standard parameter search, as described above.

# F  MORE RESULTS ON STABLE DIFFUSION

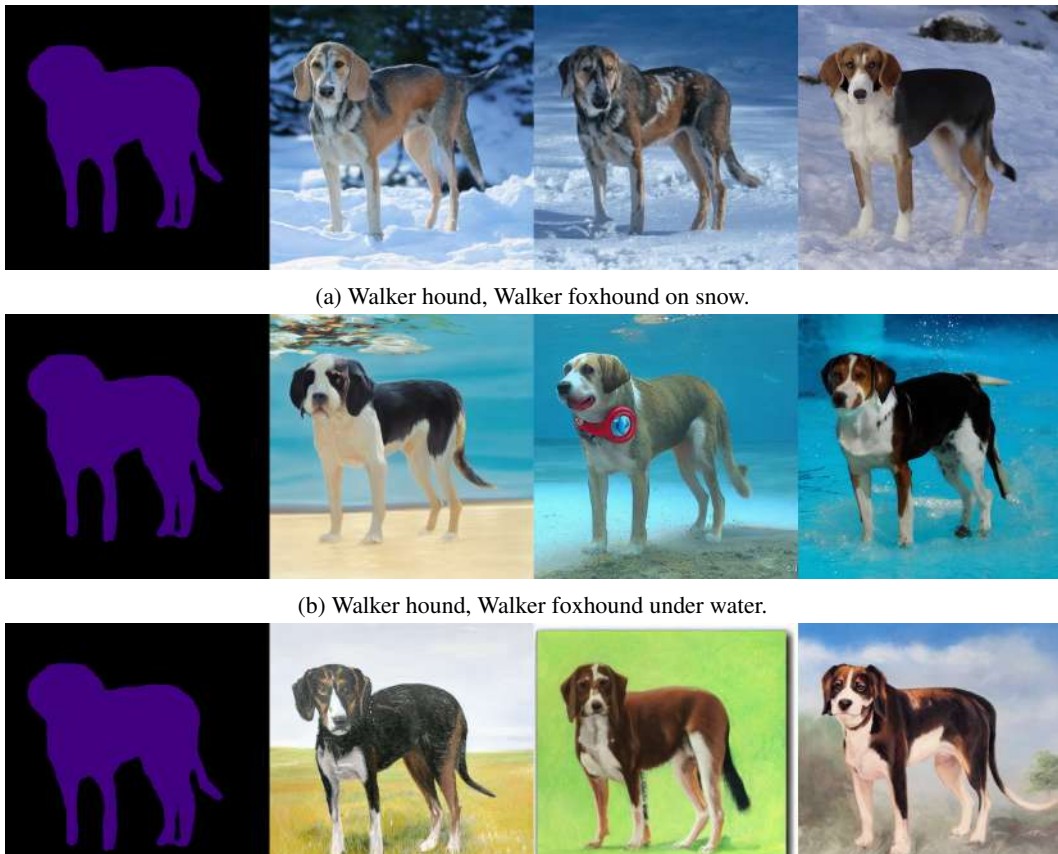

(a) Walker hound, Walker foxhound on snow.

(b) Walker hound, Walker foxhound under water.

(c) Walker hound, Walker foxhound as an oil painting.

Figure 12: More images to show Segmentation guidance. In each subfigure, the first image is the segmentation map used to guide the image generation with its caption as its text prompt.

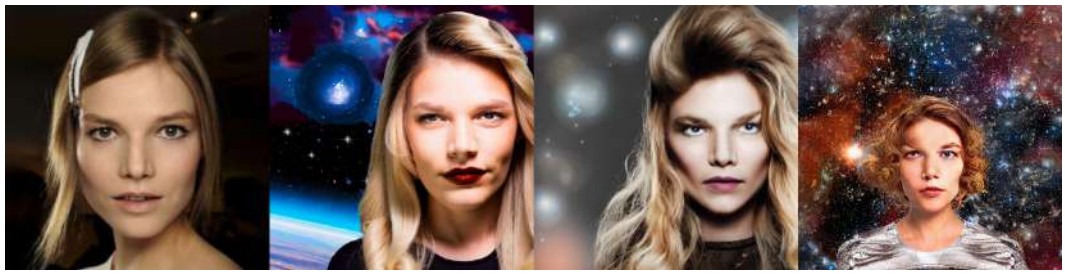

(a) Headshot of a person with blonde hair with space background.

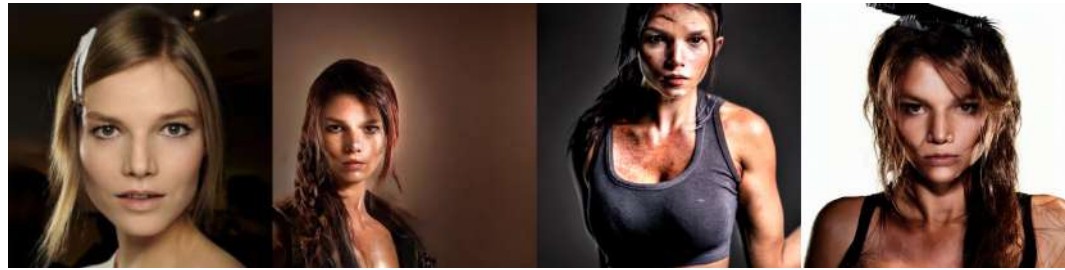

(b) A headshot of a woman looking like a lara croft.

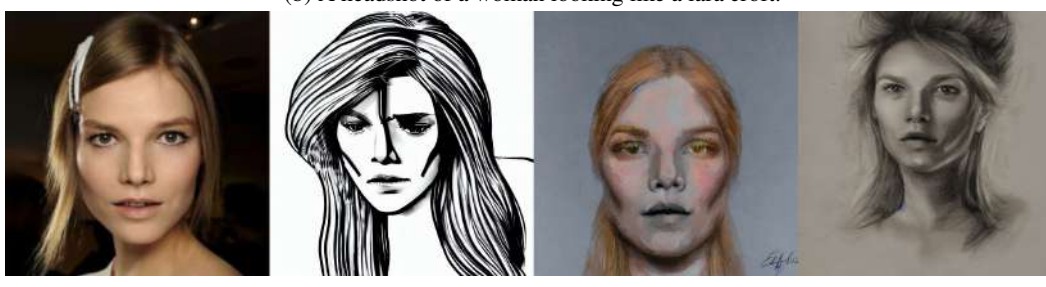

(c) A headshot of a blonde woman as a sketch.

Figure 13: More images to show Face guidance. In each subfigure, the first image is the human identity used to guide the image generation with its caption as its text prompt.

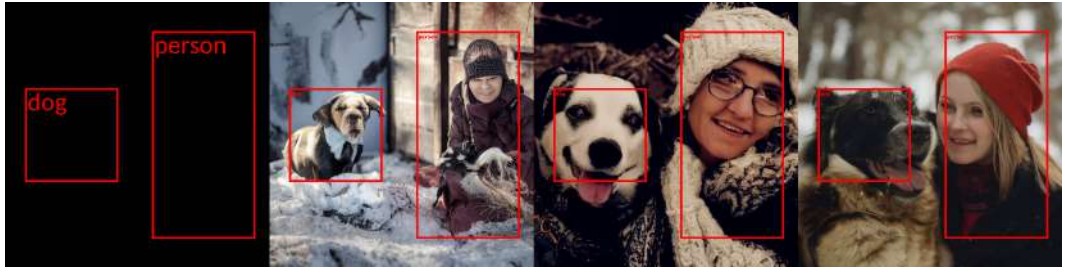

(a) A headshot of a woman with a dog in winter.

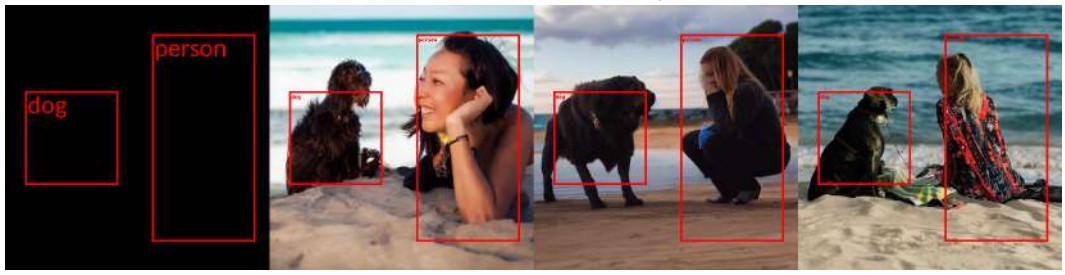

(b) a headshot of a woman with a dog on beach.

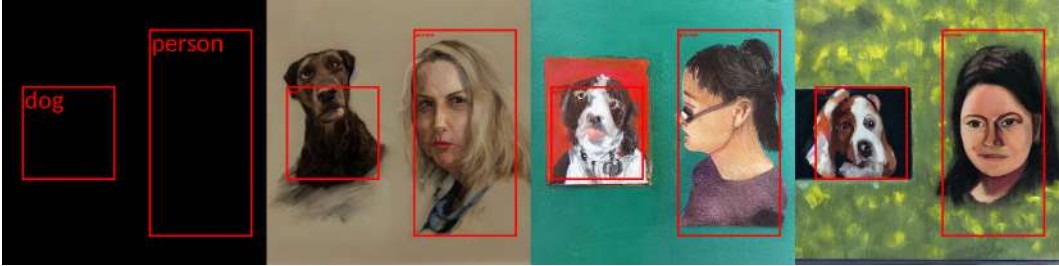

(c) An oil painting of a headshot of a woman with a dog.

Figure 14: More images to show Object Location guidance. In each subfigure, the first image is the object location used to guide the image generation with its caption as its text prompt.

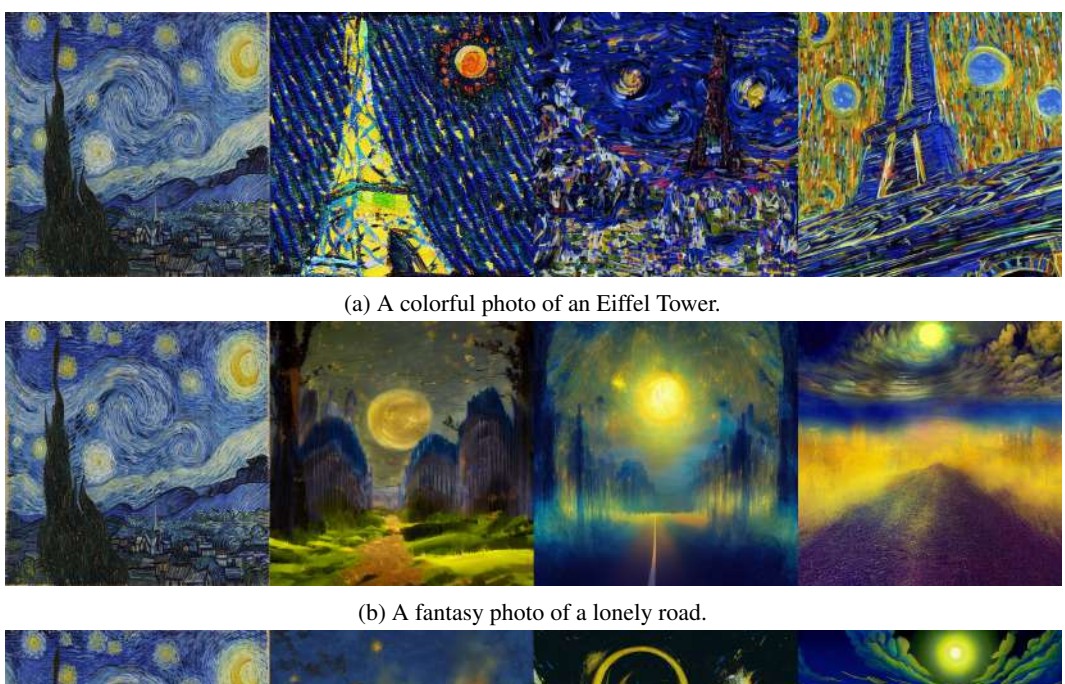

(a) A colorful photo of an Eiffel Tower.

(b) A fantasy photo of a lonely road.

(c) A fantasy photo of volcanoes.

Figure 15: More images to show Style Transfer. In each subfigure, the first image is the styling image used to guide the image generation with its caption as its text prompt.

