# OpenReview forum: "Universal Guidance for Diffusion Models"
_ICLR.cc/2024/Conference — ICLR 2024 poster_

### Official Review · Reviewer_SEzD · 2023-10-28

**Soundness:** 3 good
**Presentation:** 3 good
**Contribution:** 2 fair
**Rating:** 6
**Confidence:** 4

**Summary:**

The authors propose a novel method to control generation process by diffusion models using arbitrary guidance functions without any retraining. The proposed method comprises two types of guidance: forward guidance and backward guidance. In the forward guidance, an estimated clean data is used to compute the guidance function, and its gradient is directly employed to adjust the estimated noise at each timestep. On the other hand, in the backward guidance, the adjustment is obtained by multiple step gradient descent, which leads to more faithful results to the constraint incurred by the guidance. The exprimental results show that the proposed method works well across various types of guidance.

**Strengths:**

- The proposed method can be applied to a wide variety of conditional generation tasks without retraining the base diffusion model.

- In the experimetns, the proposed method performs well in terms of the quality of the generated images. It is also impressive that the proposed method allows for a variety of conditional generation.

- Overall, the manuscript is well-written and easy to follow.

**Weaknesses:**

<Major ones>

- The novelty in methodology is marginal.
  - In the forward guidance, the loss function is computed based on the clean data predicted from the noisy data at each timestep, and its gradient with respect to the original noisy data is used for the guidance. This approach is quite similar to that employed in DPS and FreeDoM [R1].
  - In addition, the stepwise refinement is also similar to time-travel strategy in FreeDoM.
    - [R1] "FreeDoM: Training-Free Energy-Guided Conditional Diffusion Model," ICCV 2023.

- The advantage of the proposed method over existing methods is not clear. The experiments lack comparison with baseline methods, though several related studies are refered in Section 2. Specifically, Diffusion Posterior Sampling (and FreeDoM as well) is closely related to this work and should be compared qualitatively and quantitatively.


<Minor ones>

- As any kind of guidance function can be used in the proposed method, it would be interesting to see how its performance varies according to the design of the guidance function. For example, in the case of segmentation map guidance, we may have a lot of publicly-available segmentation models that can be utilized for the guidance. In this case, can we simply choose the best performing model? This point is not clear in the manuscript, because only single model is examined for each conditional generation task.


---

<Update after the rebuttal period>

Thanks for the response. I especially appreciate the additional results on comparison with LGD. As LGD can be seen as a modified version of DPS, it becomes clear to see the advantge of the proposed method over DPS as well as LGD. Recognizing the significance of the empirical performance of the proposed method, I have revised my score from 5 to 6.

**Questions:**

Please see weaknesses.

**Details Of Ethics Concerns:**

This concern might be too much, but I would like to raise this just in case. The face recognition guidance introduced in the experiments could potentially be used to generate deepfakes of any target person if his/her facial image is available.

---

> ### Author Response · Authors · 2023-11-23
> **Individual response to Reviewer SEzD**
>
> We thank the reviewers for the effort put into reviewing this paper, and we appreciate the valuable insight on related works.
>
> > Comparison with other works
>
> Compared to our algorithm, DPS [R2] attempted to address guided generation using general guidance functions, but they proposed to apply the gradient of the $\ell_{p}$ loss between the ground-truth condition and the output of the guidance function given a predicted clean image. It is unclear how DPS can handle conditions that cannot be compared using  $\ell_{p}$ loss, such as bounding box locations, from both method perspectives. On the other hand, our forward guidance using no step-wise refinement can be viewed as a generalization of DPS that accepts any type of condition and guidance function. Please refer to Figure 10 and Section E in the appendix for comparison between whether the guided generation uses step-wise refinement or not. We also remark that, from the method perspective, we make novel technical contributions by proposing backward guidance and a step-wise refinement algorithm to further improve the results. We also empirically demonstrate the success of our algorithm on more complex guidance functions. Please also check our additional comparison with LGD, another recent work on guided image generation, in the general response. The results validate the contribution and novelty of our algorithm, especially on widely-encountered guidance functions in real world.
>
> For comparison with FreeDoM, due to a highly unusual issue, we believe the discussion should not be visible to the public. We communicate the discussion with AC via a private post and please directly contact the AC for the discussion.
>
> > Different guidance networks for a specific task
>
> In principle, our algorithm is not restrictive to the guidance function used. So we believe it is totally possible to experiment with different segmentation networks and simply pick the best one for empirical use.
>
> **Reference**
>
> [R1] "FreeDoM: Training-Free Energy-Guided Conditional Diffusion Model," ICCV 2023.
>
> [R2] Chung, Hyungjin, Jeongsol Kim, Michael T. Mccann, Marc L. Klasky, and Jong Chul Ye. "Diffusion posterior sampling for general noisy inverse problems." arXiv preprint arXiv:2209.14687 (2022).

---

### Official Review · Reviewer_xJFT · 2023-10-30

**Soundness:** 2 fair
**Presentation:** 2 fair
**Contribution:** 2 fair
**Rating:** 3
**Confidence:** 4

**Summary:**

This submission deals with designing a universal guidance for diffusion models that can adapt to various guidance modalities such as segmentation masks, bounding boxes, in a plug-and-play fashion without any retraining from the scratch or any finetuning. To solve the problem, it proposes to add a loss gradient term to the denoising score function. Let c be the guidance that is the output of some function c=f(z0) for the unknown sample z0. Then the loss is defined to be defined between c and f(\hat{z}), where \hat{z} is an estimate for the z0 based on noisy observation zt based on MMSE estimation. This approximation has been commonly used in the diffusion literature e.g., in DDIM, DPS [Chung et al’22 ], PGDM [Song et al’22]. Experiments show that this loss-guidance works to properly guide stable diffusion sampling.

[Song et al’22] Song, Jiaming, Arash Vahdat, Morteza Mardani, and Jan Kautz. "Pseudoinverse-guided diffusion models for inverse problems." In International Conference on Learning Representations. 2022.

[Chung et al’22 ] Chung, Hyungjin, Jeongsol Kim, Michael T. Mccann, Marc L. Klasky, and Jong Chul Ye. "Diffusion posterior sampling for general noisy inverse problems." arXiv preprint arXiv:2209.14687 (2022).

[Song et al’23] Song, Jiaming, Qinsheng Zhang, Hongxu Yin, Morteza Mardani, Ming-Yu Liu, Jan Kautz, Yongxin Chen, and Arash Vahdat. "Loss-Guided Diffusion Models for Plug-and-Play Controllable Generation." (2023).

**Strengths:**

Controllable generation from diffusion models without re-training or finetuning is an important problem

**Weaknesses:**

The idea in this work doesn't seem to be novel. Loss guidance for diffusion models for the same purpose has been studied in previous works that have not been cited; see [Song et al’23]. Also, the idea of using \hat{z} to approximate z0 based on the score has been used several times in the samping diffusion models for example for inverse problems as in PGDM [Song et al’22], and DPS [Chung et al’22 ].


[Song et al’23] Song, Jiaming, Qinsheng Zhang, Hongxu Yin, Morteza Mardani, Ming-Yu Liu, Jan Kautz, Yongxin Chen, and Arash Vahdat. "Loss-Guided Diffusion Models for Plug-and-Play Controllable Generation." (2023).

**Questions:**

The authors need to clarify the contributions of this work compared with previous works, especially the ones in PGDM [Song et al’22], DPS , and [Song et al’23]. I am willing to change my score if the author could clarify the contributions and differences from the work in [Song et al’23].

---

> ### Author Response · Authors · 2023-11-23
> **Individual response to Reviewer xJFT**
>
> We thank the reviewer for the effort put into reviewing this paper, and we appreciate the valuable reference the reviewer brings up. While the papers mentioned by the reviewers have explored various ways to perform loss-guided generation on diffusion models, our work exhibits critical differences and novel contributions compared to these prior arts. We discuss the difference individually for each paper mentioned by the reviewer.
>
> > Comparison with PGDM
>
> PGDM [Song et al’22] assumes the existence of a pseudo-inverse function given the guidance function, and it is unclear in the paper how their algorithm can be extended to general guidance functions. In contrast, our proposed universal guidance combines forward and backward guidance to achieve guided image generation using any guidance function, where backward guidance is a more general algorithm that reduces to PGDM if the pseudo inverse function is known.
>
> > Comparison with DPS
>
> DPS [Chung et al’22] attempted to address guided generation using general guidance functions, but they proposed to apply the gradient of the $\ell_{p}$ loss between the ground-truth condition and the output of the guidance function given a predicted clean image. It is unclear how DPS can handle conditions that cannot be compared using  $\ell_{p}$ loss, such as bounding box locations, from both method and empirical perspectives. On the other hand, our forward guidance can be viewed as a generalization that accepts any type of condition and guidance function. We also propose backward guidance and a step-wise refinement algorithm to further improve the results, and demonstrate the generation quality on complex tasks such as segmentation map guidance and object location guidance.
>
> > Comparison with [Song et al’23]
>
> The method proposed in [Song et al’23], which we abbreviate as LGD, also addresses the issue of DPS and can handle general guidance function. They further proposed to compute the guidance gradient based on the loss average over different noisy samples of the predicted clean image. Different from their work, we propose a universal refinement algorithm that refines the sampling in each diffusion step by reinjecting suitable Gaussian noise. On top of that, we also propose backward guidance that further enhances the match between the generated images and the condition. Please also check our additional comparison with LGD in the general response, which validates the contribution and novelty of our algorithm, especially on widely-encountered guidance functions empirically.
>
> > Summary
>
> In summary, our algorithm offers novel technical contributions that are significantly different from prior works by combining forward guidance, backward guidance and step-wise refinement. Our paper also offers novel empirical contributions by showing the success on more sophisticated and empirically applicable guidance functions such as segmentation and object detection networks. We hope the discussion and the experiment address the reviewer's concern about novelty and contribution, and hope that the reviewer will raise the final rating of the paper.
>
>
>
>
>
> **Reference**
>
> [Song et al’22] Song, Jiaming, Arash Vahdat, Morteza Mardani, and Jan Kautz. "Pseudoinverse-guided diffusion models for inverse problems." In International Conference on Learning Representations. 2022.
>
>
> [Chung et al’22] Chung, Hyungjin, Jeongsol Kim, Michael T. Mccann, Marc L. Klasky, and Jong Chul Ye. "Diffusion posterior sampling for general noisy inverse problems." arXiv preprint arXiv:2209.14687 (2022).
>
> [Song et al’23] Song, Jiaming, Qinsheng Zhang, Hongxu Yin, Morteza Mardani, Ming-Yu Liu, Jan Kautz, Yongxin Chen, and Arash Vahdat. "Loss-Guided Diffusion Models for Plug-and-Play Controllable Generation." (2023).

---

### Official Review · Reviewer_Dj3j · 2023-11-01

**Soundness:** 3 good
**Presentation:** 4 excellent
**Contribution:** 3 good
**Rating:** 6
**Confidence:** 3

**Summary:**

Sampling images from pretrained diffusion models using conditions is now a very hot topic and plays very important roles in many application scenarios. There are two categories of methods for conditional generation: 1) retraining with new conditions, such as Control-Net; 2) using a guidance function to optimize a latent in the pretrained image space. Considering the cost of re-training, this paper targets a universal guidance strategy for conditional sampling. The experiments demonstrate the results of many different conditions: classifier label, human identity, segmentation maps and object locations etc.

**Strengths:**

- The motivation is very clear and the writing is very easy-to-follow.
- Evenly the idea seems simple, it is novel and reasonable.
- The experiments contain many different condition scenarios and the results are visually good.

**Weaknesses:**

My major concern is on the experiments:

* It lacks ablation: as claimed in Sec 3.1, "directly replacing fcl and lce with any off-the-shelf guidance and loss functions does not work in practice". I think it is needed to include an experiment to support this claim. In addition, only "object location guidance" in Sec 4.2 provided the ablation of forward and backward guidance, the results for at least more than one tasks are helpful.

* I appreciate the experiments on conditional stable diffusion in Sec4.1 and unconditional imagenet diffusion in Sec 4.2. However, I am curious why not also conducting segmentation map guidance in 4.2. I think including same task in both 4.1 and 4.2 can be helpful for readers to understand the differences between the two diffusion models.

* My primary concern is: all presented visual results are not diverse - dogs are everywhere. I also checked all results in the supplemental, it is similar. How about other cases?

**Questions:**

See weakness part.

---

> ### Author Response · Authors · 2023-11-23
> **Individual response to Reviewer Dj3j**
>
> We thank the reviewer for the effort put into reviewing this paper, and we appreciate the insightful feedback on further ablation studies. Please check the experiments below for each individual question raised.
>
>
> > Ablation between our algorithm and directly replacing fcl and lce with off-the-shelf guidance (Eq. 5).
>
> As suggested by reviewers, we also present 8 more combinations of text and bounding boxes. We summarize the quantitative results in the table below, and present the qualitative results in Figure 1 of the [anonymized rebutall figures collection](https://drive.google.com/file/d/1DVmdqZ8QBcopO0npQLXGWdDDWQyZp3Qa/view?usp=sharing). The value is obtained by averaging over 10 random generations. We note that the "text" in the following table refers to a combination of object locations and text phrases, and please see the figure for details of each combination. Both quantitative and qualitative results demonstrate that using our proposed forward guidance with Eq. 6 leads to a much better match with the external guidance compared to Eq. 5 for off-the-shelf guidance functions.
>
> |         | text 1 | text 2 | text 3 | text 4 | text 5 | text 6 | text 7 | text 8|
> |---      | ---    | ---    | ---    | ---    | ---    | ---    | ---    | ---   |
> |Eq. 6 (Ours)|0.51 | 0.69   | 0.69   | 0.92   | 0.44   | 0.61   | 0.55| 0.80|
> |Eq. 5       | 0.13| 0.01   | 0.01   | 0.08   | 0.02   | 0.01   | 0.02| 0.06|
>
>
> > Segmentation guidance on ImageNet diffusion model
>
> As suggested by the reviewer, we also present the generation results on ImageNet diffusion model using segmentation guidance in Figure 2 and Figure 3 of the [anonymized rebuttal figures collection](https://drive.google.com/file/d/1DVmdqZ8QBcopO0npQLXGWdDDWQyZp3Qa/view?usp=sharing). In particular, Figure 2 shows images generated with various segmentation maps, demonstrating that our algorithm is effective regardless of the underlying diffusion model. Figure 3 includes comparison between generations using ImageNet diffusion model and Stable Diffusion with the same segmentation mask.
>
> > Generations using objects other than dog as conditions
>
> In Figure 1 of the [anonymized rebuttal figures collection](https://drive.google.com/file/d/1DVmdqZ8QBcopO0npQLXGWdDDWQyZp3Qa/view?usp=sharing), we show generations with various combinations of dogs and cats in different positions. The results indicate that our algorithm can generate images beyond the "dog category", and we hope this alleviates the reviewer's concern about the diversity of generated images.

---

### Official Review · Reviewer_9puQ · 2023-11-05

**Soundness:** 3 good
**Presentation:** 3 good
**Contribution:** 2 fair
**Rating:** 6
**Confidence:** 3

**Summary:**

The paper proposes a universal guidance algorithm to make the diffusion model condition on multiple modalities without retraining the model. The core idea is to use the estimated clean image in the intermediate step to provide the classifier guidance. The experiments demonstrate the effectiveness of the proposed method.

**Strengths:**

1) The proposed method enables the pretrained diffusion model to condition multiple modality controls without any retraining.

2) The paper is well-written and easy to follow.

3) The experiment demonstrates the effectiveness of the proposed method.

**Weaknesses:**

1) The method uses the predicted clean image as the input of the conditioning model. Would the inaccurate predicted clean image affect the performance?

2) Some previous work [1, 2, 3] on the conditional diffusion model should be discussed.

[1] Chong Mou, Xintao Wang, Liangbin Xie, Yanze Wu, Jian Zhang#, Zhongang Qi, Ying Shan, Xiaohu Qie. T2I-Adapter: Learning Adapters to Dig out More Controllable Ability for Text-to-Image Diffusion Models

[2] Lvmin Zhang, Anyi Rao, Maneesh Agrawala. Adding Conditional Control to Text-to-Image Diffusion Models

[3] Ziqi Huang, Kelvin C.K. Chan, Yuming Jiang, Ziwei Liu. Collaborative Diffusion for Multi-Modal Face Generation and Editing.

**Questions:**

The method uses the predicted clean image as the input of the conditioning model. Would the inaccurate predicted clean image affect the performance? The authors need to study this issue.

---

> ### Author Response · Authors · 2023-11-23
> **Individual response to Reviewer 9puQ**
>
> We thank the reviewer for the effort put into reviewing the paper and providing valuable feedback.
>
> > Would the inaccurate predicted clean image affect the performance?
>
> Indeed, as the reviewer mentioned, we use predicted clean images to compute the guidance signal and perform guided image generation. We argue that inaccurate clean image prediction has limited impact due to the step-by-step nature of image sampling for diffusion model. Specifically, impacts of guidance using gradient signal is averaged out across different sampling step. In the earlier sampling steps, the guidance signal provides coarse adjustment to the image based on comparatively inaccurate clean image predictions. As the sampling step progresses, the clean image prediction becomes more accurate and the corresponding guidance signal further refines the generated image. The success of our algorithm also validates the reasoning.
>
> > Discussion of some previous works
>
> We thank the reviewer for providing pointers to these valuable references. Collectively, these reference papers belong to a family of algorithms that freezes the unconditional diffusion backbone and train auxiliary models for controlled image generation. This family of algorithms has a different goal in mind compared to our proposed Universal Guidance, where we aim for an algorithm that does not require re-training at all for different conditional generation tasks. Nevertheless, we believe this family of algorithms is an interesting middle ground between our algorithm and full-blown conditional training,  and presents exciting directions for further research. We will include the related discussion upon further revision of the paper.

---

> > ### Comment · Reviewer_9puQ · 2023-11-23
> > **After Rebuttal Comments**
> >
> > Dear Authors,
> >
> > Thanks for addressing my concerns. I will keep my original rating for accepting this paper.
> >
> > Thanks,

---

### Author Response · Authors · 2023-11-23
**General response**

We thank all the reviewers for their efforts in reviewing the paper and providing valuable feedback. In particular, we appreciate all reviewers' agreement on the empirical importance of the research problem and our related algorithms. We also appreciate Reviewer 9puQ, Dj3j and SEzD for recognition of the writing and the empirical success of our proposed algorithm. Here we provide a general response that addresses common questions among reviewers, and please check individual responses for individual questions. We also perform various additional experiments and summarize the qualitative results in [anonymized rebuttal figures collection](https://drive.google.com/file/d/1DVmdqZ8QBcopO0npQLXGWdDDWQyZp3Qa/view?usp=sharing).


> Comparison with prior works.

We compare our algorithm with LGD [1], a recent work also focusing on guided image generation. We perform image generation guided by segmentation map and object location using Stable Diffusion. Comparison on segmentation guided sampling can be found in Figure 3 and Figure 4 of the [anonymized rebuttal figures collection](https://drive.google.com/file/d/1DVmdqZ8QBcopO0npQLXGWdDDWQyZp3Qa/view?usp=sharing), where LGD often fails to match the object with the given segmentation map. Comparison on object detection guidance can be found in Figure 5 and Figure 6 of the [anonymized rebuttal figures collection](https://drive.google.com/file/d/1DVmdqZ8QBcopO0npQLXGWdDDWQyZp3Qa/view?usp=sharing). In the comparison, LGD generated distorted images when trying to achieve better match with the given bounding boxes, indicating a failure to balance between condition match and the image realism. The comparison shows our empirical contribution on the success of more complex and widely applicable conditional generation tasks.

**Reference**

[1] Song, Jiaming, Qinsheng Zhang, Hongxu Yin, Morteza Mardani, Ming-Yu Liu, Jan Kautz, Yongxin Chen, and Arash Vahdat. "Loss-Guided Diffusion Models for Plug-and-Play Controllable Generation." (2023).

---

### Meta-Review · Area_Chair_qcZx · 2023-12-11

**Metareview:**

Summary
This work presents a method to control generation process of diffusion models using arbitrary guidance functions without any retraining. THe proposed method comprises forward guidance for generic guidance function, backward guidance for better constraint matching, and stepwise refinement to improve fidelity. The effectiveness of the proposed method is validated across various types of guidance.

Strengths:
Introduce a universal guidance method allowing for conditional generation across variouis types of guidance without retraining.
Extensive experiments demonstrate the effectiveness of the proposed method.
The paper is well-written, making it easy to follow and understand.

Weaknesses:
Concern about novelty compared to DPS, LGD and FreeDoMis is raised by some reviewers and is addressed by author through detailing difference from these methods and adding more comparison experiment.

**Justification For Why Not Higher Score:**

This work is not the first to achieve universal guidance generation. The difference and comparison with FreeDoM should be further explained and compared.

**Justification For Why Not Lower Score:**

The proposed method is effective in enabling conditional generation across variouis types of guidance without retraining and validated through abaltion study and comparison.

---

### Decision · Program_Chairs · 2024-01-16

Accept (poster)